Effects of larval foam-making and prolonged terrestriality on morphology, nitrogen excretion and development to metamorphosis in a Leptodactylid frog

http://orcid.org/0000-0001-6127-3377 Méndez-Narváez Javier 1 2 jmendez@fundacioncalima.org
http://orcid.org/0000-0002-7804-800X Warkentin Karen M. 2 3
1 Calima, Fundación para la Investigación de la Biodiversidad y Conservación en el Trópico , Cali , Colombia
2 Biology Department, Boston University , Boston, Massachusetts , United States
3 Smithsonian Tropical Research Institute , Panama , Panama
Brygadyrenko Viktor
Electronic publication date: 2025 Feb 26
Publication date: 2025
Volume: 13
Electronic Location ID: e18990
Received 2024 Oct 18; Accepted 2025 Jan 23
Copyright: © 2025 Méndez-Narváez and Warkentin
Copyright year: 2025
Copyright holder: Méndez-Narváez and Warkentin
License: This is an open access article distributed under the terms of the Creative Commons Attribution License, which permits unrestricted use, distribution, reproduction and adaptation in any medium and for any purpose provided that it is properly attributed. For attribution, the original author(s), title, publication source (PeerJ) and either DOI or URL of the article must be cited.
License URL: https://creativecommons.org/licenses/by/4.0/

Keywords: Anura, Terrestrial development, Developmental arrest, Urea excretion, Leptodactylidae, Dehydration risk, Larval period

Funding: Colombian Ministerio de Ciencia, Tecnología e Innovación (Colciencias) and Fulbright (PhD grant obtained in 2015) Smithsonian Tropical Research Institute Chicago Herpetological Society National Science Foundation IOS-1354072 Funding was provided by the Colombian Ministerio de Ciencia, Tecnología e Innovación (Colciencias) and Fulbright (PhD grant obtained in 2015), the Smithsonian Tropical Research Institute, the Chicago Herpetological Society, and the National Science Foundation (IOS-1354072). The funders had no role in study design, data collection and analysis, decision to publish, or preparation of the manuscript.

==============================
At ontogenetic transitions, animals often exhibit plastic variation in development, behavior and physiology in response to environmental conditions. Most terrestrial-breeding frogs have aquatic larval periods. Some species can extend their initial terrestrial period, as either a plastic embryonic response to balance trade-offs across environments or an enforced wait for rain that allows larvae to access aquatic habitats. Terrestrial larvae of the foam-nesting frog, Leptodactylus fragilis, can arrest development, make their own nest foam to prevent dehydration, and synthesize urea to avoid ammonia toxicity. These plastic responses enable survival during unpredictably long periods in underground nest chambers, waiting for floods to enable exit and continued development in water. However, such physiological and behavioral responses may have immediate and long-term carry-over effects across subsequent ecological and developmental transitions. We examined effects of prolonged terrestriality and larval foam-making activity on larval physiology, development, and metamorphosis in L. fragilis. We tested for changes in foam-making ability by measuring the nests larvae produced following complete removal of parental foam at different ages. We measured ammonia and urea levels in larval foam nests to assess nitrogen excretion patterns, testing for effects of larval age, soil hydration around parental nests, and repeated nest construction. We also assessed immediate and long-term effects of larval foam-making and prolonged terrestriality on larval morphology at water entry and development to metamorphosis. We found that larvae arrested development during prolonged time on land and even young larvae were able to effectively produce multiple foam nests. We found high ammonia concentrations in larval nests, very high urea excretion by developmentally arrested older larvae, and faster growth of larvae in water than while constructing nests. Nonetheless, sibling larvae had a similar aquatic larval period and size at metamorphosis, regardless of their nest-making activity and timing of water entry. Sibship size increased the size of larval foam nests, but reduced per-capita foam production and increased size at metamorphosis, suggesting maternal effects in cooperative groups. Metamorph size also decreased with aquatic larval period. Our results highlight the extent of larval ability to maintain and construct a suitable developmental environment and excrete N-waste as urea, which are both crucial for survival during enforced extensions of terrestriality. Our results suggest that the energetic reserves in large eggs are sufficient to meet metabolic costs of urea synthesis and foam production during developmental arrest over an extended period on land, with no apparent carry-over effects on fitness-relevant traits at metamorphosis.

Introduction

Most animals with complex life cycles experience ontogenetic transitions that allow them to exploit multiple environments (Truman, 2019; Laudet, 2011; Wassersug, 1975). Different life stages of these animals may experience environment-specific selective factors that can constrain or increase phenotypic diversity within each stage or between stages (Bardua et al., 2021; Gomez-Mestre & Buchholz, 2006; Phung et al., 2020). Studying how environmental conditions affect phenotypic responses across development could clarify the role of developmental plasticity in ontogenetic switches in ecology and morphology (Gilbert, 2012; West-Eberhard, 2003; Moore & Martin, 2019). For instance, early life stages of vertebrates can respond to multiple environmental conditions and cues (e.g., threats, resources) with morphological, behavioral and physiological changes at life history transition points such as hatching (Martin, 1999; Mueller et al., 2019; Warkentin, 2011) and metamorphosis (Werner, 1986). These plastic responses at a particular life stage may have immediate and long-term carry-over effects on expressed traits and survival in subsequent stages or after transitions into a new environment (Gomez-Mestre & Buchholz, 2006; Touchon & Warkentin, 2010; Cabrera-Guzmán et al., 2013; Scott et al., 2007).

The complex life cycles of anurans are characterized by multiple ecological and physiological shifts between aquatic and terrestrial environments (Crump, 2015; Gomez-Mestre, Pyron & Wiens, 2012). Many studies have assessed how environmentally induced responses in the aquatic larval environment (Gomez-Mestre et al., 2010; Laurila & Kujasalo, 1999; Vonesh & Warkentin, 2006) influence traits in subsequent life stages, including nutrition uptake efficiency and growth (Bouchard et al., 2015; Zhu et al., 2019) with its underlying physiological pathways (Burraco, Díaz-Paniagua & Gomez-Mestre, 2017; Crespi & Warne, 2013; Murillo-Rincón, Laurila & Orizaola, 2017). Emerging metamorphs and juveniles carry effects of larval nutrition and physiology that contribute to variation in locomotor performance, behavior, and survival on land (Bouchard et al., 2016; Gomez-Mestre et al., 2010; Tarvin et al., 2015). While a large body of literature addresses effects of the aquatic larval environment on metamorphic and post-metamorphic phenotypes, much less is known about potential effects of terrestrial embryonic development on subsequent stages, including the aquatic larval period and metamorphosis (Ortega-Chinchilla et al., 2019; Touchon et al., 2013; Touchon & Warkentin, 2010).

Terrestrial and semi-terrestrial development are widespread and have evolved independently many times in amphibians (Gomez-Mestre, Pyron & Wiens, 2012; Liedtke, Wiens & Gomez-Mestre, 2022). Embryos and larvae have evolved their own adaptations to conditions common during terrestrial development (Delia, Ramírez-Bautista & Summers, 2014; Delia et al., 2019; Méndez-Narváez & Warkentin, 2022; Salica, Vonesh & Warkentin, 2017; Seymour & Loveridge, 1994; Shoemaker & McClanahan, 1973; Warkentin, 1995, 2007), including the ability for extended or arrested embryonic development on land (Bradford & Seymour, 1985; Moravek & Martin, 2011). In most frogs, terrestrial early development is followed by an aquatic larval period that ends in metamorphosis (Liedtke, Wiens & Gomez-Mestre, 2022). This generates life cycles with two habitat transitions separated by variable periods of growth and development within each habitat. The environmental conditions experienced in early life stages can affect hatchling size, morphology, and subsequent development and survival in the aquatic environment (Delia et al., 2019; Touchon & Warkentin, 2010; Warkentin, 1995; Willink et al., 2014). Indeed carry-over effects of terrestrial embryonic environments may still be evident at metamorphosis and in post-metamorphic stages, after frogs re-emerge onto land (Touchon et al., 2013; Touchon & Warkentin, 2010; Vonesh & Bolker, 2005). Because adaptations that facilitate terrestrial development and survival under stressful conditions may involve changes in energetic demands (Méndez-Narváez & Warkentin, 2022; Seymour, Geiser & Bradford, 1991), they are likely to alter some physiological costs and nutritional demands that affect aquatic larval development and metamorphosis (Burraco, Díaz-Paniagua & Gomez-Mestre, 2017; Crespi & Warne, 2013).

In organisms with complex life cycles, nutritional and neuroendocrine factors regulate and constrain transitions between life stages (Callier & Nijhout, 2011; Pfennig, Mabry & Orange, 1991). Indeed, during frog metamorphosis, when metabolic reorganization occurs, differentiation and growth of new structures depends on energy reserves accumulated during larval development (e.g., fat bodies and liver) (Mueller et al., 2012; Zhu et al., 2020). Urea excretion is considered a key physiological adaptation that allowed tetrapod invasion of land (Amemiya et al., 2013), but with a higher metabolic cost of synthesis in comparison with ammonia excretion (Shambaugh, 1977). In amphibians, it plays a key role in enabling the transition from aquatic to terrestrial environments, with an onset or upregulation at frog metamorphosis (Brown, Brown & Cohen, 1959; Wright & Wright, 1996; Zhu et al., 2020). However, long before metamorphosis, some terrestrial frog embryos and larvae also excrete urea (Alcocer et al., 1992; Grafe, Kaminsky & Linsenmair, 2005; Martin & Cooper, 1972; Méndez-Narváez & Warkentin, 2022; Shoemaker & McClanahan, 1973). This occurs via early onset of expression of the urea cycle enzymes arginase and CPSase (Méndez-Narváez & Warkentin, 2023) and is clearly beneficial for terrestrial survival, by preventing ammonia toxicity (Méndez-Narváez & Warkentin, 2022). However, the benefits of urea excretion for terrestrial early life stages must be balanced against the metabolic cost of urea synthesis and related traits that enable facultative extensions of terrestriality, particularly for species that do not feed before their transition to water.

We studied key larval traits that facilitate prolonged terrestrial life and their carry-over effects to metamorphosis in the white-lipped frog, Leptodactylis fragilis (Brocchi, 1877); an earlier version of this work formed part of Méndez-Narváez’ PhD dissertation (Méndez-Narváez, 2022). This species belongs to the L. fuscus group, in which adults exhibit foam-nesting behavior and larvae can produce a new foam nest or aggregate in groups for several days to survive out of water (Downie & Smith, 2003; Venturelli, Da Silva & Giaretta, 2021). In L. fragilis, early development occurs in a terrestrial foam nest, within a chamber that a male excavates near a temporary pond (Fig. 1A). Larvae must remain in the chamber until rainfall floods it, enabling them to swim out and reach nearby pools to continue development to metamorphosis (Méndez-Narváez & Warkentin, 2022). However, the volume of parental foam typically decreases over time, especially in dry soil, and it can also be dissolved by rain that is insufficient to free larvae from their chamber (Caldwell & Lopez, 1989; Downie, 1984; Méndez-Narváez & Warkentin, 2022). In this context, terrestrial larvae of L. fragilis are able to make foam to supplement or replace deteriorating parental foam or build an entirely new larval foam nest to extend their survival on land (Méndez-Narváez, 2022; Fig. 1B). In the absence of rain, nest-dwelling L. fragilis larvae appear to arrest development at Gosner (1960) stage 28, after 7 days on land, and can survive on land in this stage for several days (N Belduque-Correa, KM Warkentin & J Méndez-Narváez, 2019, personal observations). Developmental arrest has been hypothesized to reduce metabolism to conserve energy until larvae can enter water, as in the closely related L. fuscus (Downie, 1994). However, early larvae of L. fragilis can sustain periods of high activity during foam-making (Méndez-Narváez, 2022).

Figure 1 Larval foam-making in Leptodactylus fragilis: natural context and methods to assess effects on development to metamorphosis.

Adults of L. fragilis construct a foam nest in a subterranean chamber (A), where nest-dwelling larvae may spend a prolonged period waiting for rain and constructing new larval foam (B). We removed larvae from the parental nest before, near, and several days after the onset of developmental arrest (4.5 d, 8.5 d and 12.5 d, respectively) to assess their foam-making ability (C). We transferred sibling groups to small Petri dishes where we gave the larvae two days to create a new foam nest (larval nest 1), then repeated this procedure twice more (larval nests 2 and 3). To assess carry-over effects of extended terrestriality and larval foam-making, we transferred two individuals to water at 12.5 d when we moved their siblings to the nest-making treatment. We collected siblings from the water and first nest at 14.5 d, and from the third nest at 18.5 d and photographed them in three views to assess larval morphology (D). We left a non-nesting sibling in water (since 12.5 d) and transferred two from larval nests 1 (at 14.5 d) and 3 (at 18.5 d) to water to rear to metamorphosis, then assessed time and size at forelimb emergence (Gosner stage 42) and tail resorption (stage 46).

Variation in rainfall, affecting both soil hydration and when larvae can enter the water, as well as variation in the initial physical size of parental foam nests and number of larvae they contain, generates substantial variation in the need for larvae to make their own foam. We simulated this natural variation to assess the foam-making ability of larvae at different ages, before and during developmental arrest, and the short- and long-term consequences of prolonged terrestriality and larval foam-making activity on physiological and developmental traits. (1) Because foam-making depends on the development and function of foamy glands and performance of foam-making behavior (Giaretta et al., 2011), and because natural selection for foam-making may intensify as parental foam deteriorates over time (Méndez-Narváez, 2022), we hypothesized that foam-making ability increases with development or age. (2) Because foam-making appears energetically costly and metabolic activity generates waste-products, which would accumulate in an initially small amount of foam after nest loss (Méndez-Narváez, 2022), we hypothesized that constructing new larval foam nests also requires increased synthesis of less-toxic urea to avoid ammonia toxicity, thereby increasing energetic costs. (3) Because yolk reserves are the sole, limited energy source for these terrestrial larvae (Womble, Pickett & Nascone-Yoder, 2016), we hypothesized that larval foam-making, especially repeated nest construction, may deplete their energy reserves or trade-off with growth to affect size or morphology when they enter the water, with potential carry-over effects on traits at metamorphosis.

Materials and Methods

Study site and experimental conditions

We conducted field work during the rainy season, from May to October of 2017 and 2018, in Gamboa, Panamá (9°07′14.8″N, 79°42′15.4″W) with permission from the Panamanian Ministry of the Environment (MiAmbiente permits SE/A-56-17, SC/A-51-18) and approval from the Animal Care and Use Committee of the Smithsonian Tropical Research Institute (STRI–ACUC protocol # 2016-0520-2019A1–A3). We identified and marked the burrows made by male L. fragilis, adjacent to ephemeral pools in their territories, and checked them daily. When we found a new foam nest in a previously empty chamber, the morning after oviposition, we removed its cover (soil or dead vegetation, Fig. 1A), extracted the nest with a moistened spoon, and placed it in a small Petri dish, then hand carried it in a plastic cooler to a nearby open-air laboratory (~26 °C, ~85% RH). We buried each nest individually in an artificial burrow within a plastic container (24 cm diameter × 14 cm height) filled with soil collected near the breeding sites. For most nests, we matched soil water content in the laboratory to that found in the field, spraying the soil with water once daily to maintain hydration (wet conditions). We kept a subset of nests under reduced soil water content (~50% lower, dry conditions) to simulate a period without rainfall, following methods in Méndez-Narváez & Warkentin (2022), to assess effects of initial soil hydration on nitrogen wastes in new foam nests made by larvae.

Larval foam-making ability across ages

To assess changes in the ability of L. fragilis larvae to construct new foam nests during a prolonged time on land, we tested them at Gosner stage 24 (4.5 days), and Gosner stage 28 (8.5 and 12.5 d), i.e., before, near, and several days after the onset of developmental arrest (N Belduque-Correa, KM Warkentin & J Méndez-Narváez, 2019, personal observations). For each sibship, we maintained embryos, then larvae, in their original (parental) foam nest in wet soil until testing, then dissolved the nest with aged tap water. We counted the larvae and used a plastic transfer pipette to move them to a small Petri dish (60 mm diameter × 15 mm depth). We kept all sibling larvae together in aged tap water for 2–4 h before draining their dish (original group size 39–127, mean 79.0 ± 19.8 SD, N = 44). This simulates a limited flooding scenario, where rain dissolves the nest but is insufficient for larvae to move to a pond, trapping them in foamless chambers. Because larvae building nests (Fig. 1B) in this context are typically surrounded by most or all of their siblings (Caldwell & Lopez, 1989; J Méndez-Narváez, 2017, personal observations), we did not split sibships to standardize size. We left each group of larvae (N = 44 sibships) in their drained dish for 2 days, placing it vertically so tadpoles remained together at the bottom, as in a nest chamber (Fig. 1C, Video S1). Although larvae usually completed a new nest within 24 h of draining the dish, then became inactive (J Méndez-Narváez, 2017, personal observation), we allowed them 2 days to ensure nest completion. Then, to quantify the volume of foam produced, we photographed the flat, vertical face of the Petri dish with a Canon PowerShot SX40HS camera, including a scale in the frame (Fig. 1C). We used NIH ImageJ 2.0 (https://imagej.nih.gov/ij/index.html) to measure the area of foam three times from each photograph, averaged these values, and multiplied by the depth of the dish (15 mm) to calculate foam volume. To assess larval ability to repeat nest construction, which could increase metabolic costs during a prolonged non-feeding time on land, we repeated the nest dissolution and dish-draining procedure twice more, inducing groups of larvae to make second (N = 29) and third nests (N = 20). We removed two or three larvae at each nest attempt to assess immediate effects on larval morphology, and reared others to metamorphosis to assess longer-term consequences (see Carry-over effects, below).

Nitrogen excretion in larval nests during prolonged terrestriality

We used a different set of sibships to measure nitrogen wastes accumulated in larval foam nests at two stages. We induced larval groups (group size 52–119, mean 84 ± 16.12 SD, N = 33) to make new nests in Petri dishes as above, then collected the foam to assess ammonia and urea concentration. We collected foam from first larval nests made after 4.5 d in parental nests in wet soil (Gosner stage 24, N = 6 sibships), and from first and third larval nests made after 12.5 d in parental nests in wet and dry soil (Gosner stage 28; wet: N = 12 first, 4 third nests; dry: N = 9 first, 2 third nests). To quantify nitrogenous wastes in larval nests, we followed methods previously used for L. fragilis parental foam nests (Méndez-Narváez & Warkentin, 2022). We used 3 ml plastic transfer pipettes to collect all of the larval foam from Petri dishes and stored these samples in micro-centrifuge tubes at –20 °C. For analysis, we thawed samples and centrifuged foam for five minutes at 12,000 rpm to obtain the liquid portion for enzymatic quantification of ammonia and urea with a commercial kit (Boehringer Mannheim Cat. No. 10542946035). We assessed ammonia and urea concentration simultaneously, using 0.1 ml of the sample for each, in two quartz cuvettes, and measuring changes in absorbance at 340 nm at room temperature with a UV-Visible Spectrophotometer (Thermo-Scientific Evolution 60S). We ran controls without samples to assess background absorbance of kit reagents. Some samples produced values below the detection limit (0.47 and 0.23 mmol/L for ammonia and urea; reported as zeros) or absorbances too high to quantify concentrations, reported as NA, when N-wastes measured above the quantifiable range (>4.79 and 2.33 mmol/L for ammonia and urea) and there was insufficient sample to dilute and repeat the analysis (Méndez-Narváez & Warkentin, 2022). For each sample, we also calculated total N-waste accumulation as [ammonia] + 2 * [urea]; assuming all urea was produced by conversion of ammonia, this “predicted concentration” of ammonia estimates what could have accumulated in nests without urea excretion. For first nests only, we estimated the amount (μmol) of ammonia, urea, and predicted ammonia excreted per larva into their new foam, from the recovered liquid portion of foam (waste mass = concentration × liquid volume, i.e., assuming no losses or other sources of these molecules) and compared their excretion following development in parental nests in wet vs. dry soil.

In addition, to measure tissue urea levels and activity of the enzyme arginase, which produces urea in the ornithine urea cycle (Brown, Brown & Cohen, 1959), we collected larval sibships that had made one new nest in a Petri dish, as above, after 12.5 d in their parental nest in wet or dry soil (Gosner stage 28, N = 7 and 5 sibships, respectively). We followed methods previously used for successful quantification of arginase activity and urea in tissues of terrestrial L. fragilis larvae from parental foam nests (Méndez-Narváez & Warkentin, 2023). Briefly, we snap froze all sibling larvae from a Petri dish in liquid nitrogen and stored them, intact, in a micro-centrifuge tube at –80 °C until biochemical quantification of enzymatic activity. We quantified arginase activity (μmol min−1 mg−1 of protein) by a colorimetric method (Felskie, Anderson & Wright, 1998) and urea in tissues (μmol mg−1 of wet mass) with the diacetyl-monoxime method (Rahmatullah & Boyde, 1980), using the biochemical conversion of arginine to urea. We pooled all larvae in a sibship and ground frozen specimens to a fine powder in a mortar, using a pestle and liquid nitrogen, then prepared extracts by homogenizing a sample of ~50 mg of tissue, with 4 volumes of homogenization buffer (20 mM K2HPO4, 10 mM Hepes buffer, pH 7.5, 0.5 mM EDTA, 1 mM DTT). We determined protein concentration in each sample by the dye-binding method (Bradford, 1976) with the Thermo Scientific Coonassie Protein Assay kit. We performed all enzymatic assays at 26 °C, from fresh homogenates stored at −80 °C for not more than 1 month after homogenization. Enzymatic activities were standardized to specific activity using the protein concentration in each sample (μmol min−1 mg−1 of protein). The estimated detection limit for arginase activity was 0.001 μmol min−1 mg−1 of protein.

Carry-over effects of prolonged terrestriality and larval foam-making

To assess carry-over effects of larval foam-making activity, coupled with the extended terrestriality that necessitates it, we used larvae from the “foam-making ability” experiment (above). From 14 sibships removed from parental nests after 12.5 d (Gosner stage 28), we moved two or three larvae from the Petri dish to 150 ml of water in a plastic cup before draining the dish to induce their siblings to make foam. Like the larvae making foam, larvae in water were not fed, thus depended on yolk reserves. At 14.5 d, we collected two larvae from the first larval nest and one sibling from the water to compare their morphology (Fig. 1D, N = 13 sibships). We collected two additional larvae per sibship from their second and third nests (at 16.5 and 18.5 d, N = 14 sibships) to assess the effects of repeated construction of larval foam nests on larval phenotypes. We sacrificed larvae by immersion in an overdose of the anesthetic MS-222 (tricaine methane sulfonate) at 250 g/L, buffered with sodium bicarbonate, and preserved them using buffered 10% formalin. Within 3 months of preservation, we staged each specimen under a dissecting microscope (Zeiss Stemi DV4 Stereo), following Gosner (1960); all were at stage 28. We took dorsal, lateral and ventral photographs with a Canon EOS 5D Mark III, including a scale and used NIH ImageJ 2.0 to measure the seven linear dimensions most commonly used to compare tadpole morphology (McDiarmid & Altig, 1999) from these images. We measured total length (TL), tail length (TAL), tail muscle width (TMW), interorbital distance (IOD) and head width (HW) in dorsal view, and tail muscle height (TMH) and tail height (TH) in lateral view. We averaged measurements across siblings at each treatment level to conduct morphometric analysis. We assessed measurement repeatability by measuring photographs in triplicate for a random subset of 29 individuals, across sibships and treatment levels, and assessing the coefficient of variation for each measurement (mean CV = 1.22%, Table S1).

To assess long-term carry-over effects, we reared larvae to metamorphosis (Fig. 1E). These included one or two larvae per sibship that made no new nest, entering the water at 12.5 d (from 25 sibships) plus four per sibship that made foam for one or three larval nests, entering the water at 14.5 and 18.5 d (2 per age) Starting at 14.5 d, we fed aquatic larvae with rabbit chow ad libitum, replacing the food and about 80% of the water every second day. When larvae approached metamorphosis, we placed a small rock in their cup for metamorphs to climb, to avoid drowning, and began checking them daily for forelimb emergence (Gosner stage 42). At stage 42, we weighed individuals to the nearest 0.1 mg with an electronic balance, photographed them in dorsal view with a scale, and measured their total length (TL) with ImageJ. Then we ceased providing food, reduced water level to about 1 cm to prevent drowning, and checked metamorphs daily for complete tail resorption (Gosner stage 46). At stage 46, we measured their snout-vent length (SVL) to the nearest 0.1 mm with calipers and weighed them again. We compared the periods from entry into the water until forelimb emergence (henceforth “aquatic larval period”) and until complete tail resorption (“aquatic larval + metamorphic period”). We also compared size and age (measured from oviposition) at forelimb emergence and tail resorption. Froglets were kept in their plastic cup, leaving a few millimeters of water to maintain hydration, until release at their original collection site that night or the following night.

Statistical analysis

All statistical analyses were conducted in RStudio (version 1.1.463), and all model residuals were inspected for normality and homogeneity with the package DHARMa with 1,000 simulations (Hartig, 2024) to ensure data met analytic assumptions. We used linear mixed effects models (LMEM) (Bates et al., 2015; lme4) to test for differences in the volume of foam larvae produced after loss of their parental foam at three ages, with larval nest number (1, 2, 3) and size of larval group as covariables and sibship as a random factor.

We used a linear model to compare ammonia and urea concentration in new larval foam nests made after 12.5 d in parental nests on wet vs. dry soil, and in the first vs. third larval foam nest. For first foam nests only, we assessed the effect of prior soil moisture on ammonia and urea concentration with the volume of liquid in the new nest as a covariable. We compared the amount of ammonia, urea, and total N-waste excreted per larva in first larval nests across soil hydration treatments, and we tested if the amount of urea excreted per larva varied with the predicted ammonia concentration, that could have accumulated in the new larval nest. We also tested for effects of soil hydration on subsequent proportion of N excreted as urea (i.e., urea-N/total-N-waste) using a generalized linear mixed model (GLMMs) with an underlying Beta error distribution (Brooks et al., 2024; glmmTMB) and likelihood ratio test (LRT) to obtain p-values. We did not statistically analyze N-wastes for the first larval nest after 4.5 d in the parental nest, as it was measured in only a few sibships, all from wet soil (N = 6); however, we include key descriptive statistics to compare with other ages in Table 1.

Table 1 Nitrogen excretion in larval foam nests of Leptodactylus fragilis and arginase activity in larval tissue.

Time (d) and conditions in parental nest	Larval nest	Sampling age (d)	A. Concentrations of N-wastes (mmol/L)
mean ± SD, N	
Ammonia	Urea	Total N-waste	
4.5, wet	First	6.5	126.4 ± 86.3, 6	NA: 1, 0: 5	126.4 ± 86.3, 5
NA: 1	
12.5, wet	First	14.5	91.3 ± 65.1, 12	124.7± 83.9, 9
NA: 2, 0: 1	209.2 ± 131.2, 10
NA: 2	
12.5, dry	First	14.5	195.8 ±150.1, 9	461.6 ± 510.4, 9	657.3 ± 633.0, 9	
12.5, wet	Third	18.5	178.2 ± 33.5, 4	506.4 ± 105.4, 3
NA: 1	682.7 ± 120.5, 3
NA: 1	
12.5, dry	Third	18.5	191.1± 45.9, 2	598.2 ± 354.6, 2	789.3 ± 400.6, 2	
			B. Amounts (μmol) of N-wastes per larva	
12.5, wet	First	14.5	0.02 ± 0.02, 12	0.03 ± 0.03, 9
NA: 2, 0: 1	0.06 ± 0.04, 10
NA: 2	
12.5, dry	First	14.5	0.03 ± 0.02, 9	0.08 ± 0.07, 9	0.11 ± 0.09, 9	
			C. Arginase activity and urea in larval tissue	
			Arginase (μmol min−1 mg−1 protein)	Urea (μmol mg−1 wet mass)	
12.5, wet	First	14.5	0.08 ± 0.03, 7	0.22 ± 0.03, 7	
12.5, dry	First	14.5	0.11 ± 0.04, 5	0.22 ± 01, 5	
Note:

(A) Ammonia, urea, and total N-waste accumulated in foam nests made by L. fragilis larvae after 4.5 in parental foam nests on wet soil or 12.5 d in parental nests on wet or dry soil and, for older larvae, in first and third larval nests. (B) N-wastes accumulated per larva over 2 days constructing their first new nest. (C) Activity of the urea cycle enzyme arginase and concentration of urea in larval tissue. Data are mean ± SD and N for measurable values. If any values were above the measurable range (“NA”) or below the detection limit (“0”) we also report their N; all N are in bold.

We tested for short-term effects of foam-making behavior and extended terrestriality on larval morphology in two ways: by comparison of 14.5 d larvae after 2 d making foam vs. 2 d in water, and by comparison of terrestrial larvae after making 1–3 nests (starting at age 12.5 d). We first used principal component analyses (PCA) to summarize the morphological variation across larvae measured for each comparison (Kassambara & Mundt, 2020; factoextra). Then, we compared principal component scores (PC1 to PC3) for each analysis with LMEM, including sibship as a random factor.

We tested for long-term effects of foam-making activity (0, 1, or 3 larval nests) on the age (from terrestrial oviposition) and aquatic larval period (from water entry) to forelimb emergence (GS 42), and the age and aquatic larval plus metamorphic period to tail resorption (GS 46). Then, we tested for effects of foam making, age, and the relevant aquatic period (larval or larval + metamorphic) on measures of length (total length and SVL). We also tested for effects of foam-making, age, and aquatic period on mass at each transition, including length (total or SVL) as a covariable. We used LMEM and a model selection approach using AICcWt (weighted and corrected for small sample size) to compare models (deltaAICc) and choose the best one for each test (Mazerolle, 2023; AICcmodavg). For the best models, we used ratio tests (LRT) to obtain p-values with a nested approach, removing some interactions between predictors when non-significant to estimate the main effects. We also calculated a conditional and marginal coefficient of determination, pseudo-R-squared, using MuMIn package (Barton, 2024). We made pairwise comparisons (Tukey method) using the corresponding model structure in each case (Hothorn, Bretz & Westfall, 2008, multcomp).

Results

Larval foam-making ability

Terrestrial larvae of L. fragilis were able to construct new foam nests at all tested ages (Fig. 2, Table S2), and there was no mortality during initial or repeated nest construction. Larvae actively blow bubbles to create new foam (Video S2) and we have not observed new bubbles forming another way. The total volume of foam produced by groups of sibling larvae increased with the number of individuals in a group (X2 = 22.39, p < 0.0001), regardless of the age at transfer from their parental foam (X2 = 0.40, Fig. 2A) or how many nests the larvae had made (X2 = 1.33, p = 0.51, Fig. 2B). The foam volume per larva decreased with sibship size (X2 = 15.92, p < 0.0001), with an interaction effect with age at transfer (X2 = 11.73, p = 0.003, Fig. 2C) but no main age at transfer effect (X2 = 4.72, p = 0.094). In smaller sibships, each larva made more foam, with the strongest effect for those that were oldest at transfer (12.5 d, Fig. 2C). However, the foam volume produced per larva (Fig. 2D) did not vary with how many nests the larvae had made (X2 = 2.12, p = 0.35) or the age at transfer (X2 = 4.55, p = 0.10).

Figure 2 Foam-nesting performance of Leptodactylus fragilis larvae in relation to age, sibship size, and repeated nest-making.

Foam volume produced by sibling groups over two days in a Petri dish, after removal of their prior nest. Total volume of foam per nest produced (A) after larvae developed for 4.5 d (green), 8.5 d (orange) and 12.5 d (gray) in parental foam nests in soil, pooled across larval nests 1–3, and (B) in the first (black), second (brown) and third (purple) foam nest that larvae made, pooled across age at transfer. (C) Volume of foam produced per larva in relation to sibship size at the three transferred ages, pooled across larval nest number. (D) Volume of foam produced per larva after removal from the parental nest at three ages and construction of different numbers of larval nests. Box plots show median, first and third quartiles, and extent of data to 1.5 X IQR; data points are also shown. Sample sizes (N) are indicated.

Nitrogen excretion in larval foam nests

Ammonia was detected in larval foam nests at all tested ages, even shortly after hatching (age 4.5 d; Fig. 3A, Table 1), but urea was above the detection limit only after further development on land (by age 12.5 d; Table 1, Fig. 3B). In the first nest that larvae constructed after 12.5 d in parental foam, ammonia concentration was higher if the parental nest had been on dry soil (F1,24= 4.71, p = 0.04, Fig. 3A). Ammonia concentration in larval foam was not affected by the number of larval nests constructed (first vs. third, F1,24 = 1.37, p = 0.25, Fig. 3A). Urea concentration in the new larval foam (first and third nests) was higher when individuals came from dry soil (F1,21 = 6.99, p = 0.02, Fig. 3B) and in the third vs. first larval nest (F1,21 = 11.74, p = 0.002, Fig. 3B), with no significant interaction effect.

Figure 3 Nitrogen waste accumulation in foam nests constructed by Leptodactylus fragilis larvae.

(A) Ammonia and (B) urea concentration in larval foam nests constructed after transfer from parental nests at two ages (4.5 d and 12.5 days). For older larvae, parental nests were maintained in wet (blue) or dry (brown) soil, and larvae made one or three foam nests. (C) Concentration of nitrogen wastes in relation to water volume in the larval nest and (D) the amount of ammonia and urea accumulated per larva, as well as the ammonia calculated to accumulate if none were converted to urea, is shown for first nests made by 12.5 d larvae from each soil hydration history. Box plots show median, first and third quartiles, and extent of data to 1.5 X IQR; data points are also shown. Horizontal dashed gray line shows the detection limit for urea. Sample sizes (N) are indicated.

Ammonia and urea concentrations decreased as the volume of water in first larval foam nests increased (ammonia, t = −2.77, p = 0.01; urea: t = −2.41, p = 0.03, Fig. 3C), with the volume being lower if the parental nest had been on dry soil (t = −5.28, p < 0.0001, Fig. 3C). Dry vs. wet conditions in parental nests did not affect the estimated amount of ammonia accumulated in new larval foam nests (total: t16.87 = −0.79, p = 0.40; per larva: t15.96 = −1.08, p = 0.295, Fig. 3D, Table 1), nor did they significantly affect urea accumulation (total: t10.10 = −1.85, p = 0.09; per larva: t10.31 = −1.64, p = 0.13, Fig. 3D). However, the predicted ammonia concentration in the absence of the urea cycle explained urea excretion (total: F1,15 = 140.08, p < 0.00001; individual: F1,14 = 133.43, p < 0.0001), with no interaction effect with treatment. Parental nest hydration did not affect the total N-waste accumulated in larval nests (concentration: t12.28 = −1.39, p = 0.18; amount per larva: t11.98 = −1.56, p = 0.14, Fig. 3D), nor did it affect the proportion of total nitrogen excreted as urea (wet: 0.41 ± 0.20; dry: 0.49 ± 0.11, X2 = 1.87, p = 0.17). Arginase activity (µmol min−1 mg−1 of protein) was detected in all larval tissues and did not change with parental nest hydration (t7.55 = −1.37, p = 0.21) nor did the urea concentration (μmol mg−1) in tissues (t7.55 = −0.69, p = 0.51, Table 1).

Short-term effects on larval size and morphology

Comparing the morphology of larvae at 14.5 d, after 2 d either in water or constructing a new larval foam nest, PC1, 2, and 3 accounted for 69.3%, 13.0% and 9.6% of variance, respectively (Fig. 4A, Table S3). Five measurements of body size made important contributions to PC1, all with positive loadings, with total length loading most heavily (Table S4). Tail muscle height and tail muscle width made the largest contributions to PC2 and PC3 respectively, with positive loadings (Table S4). Overall, there were morphometric differences between the larvae in these two groups (Manova, F3,22 = 9.68, p = 0.0003, Pillai’s Trace1,24 = 0.57). At 14.5 d, after making a foam nest larvae were smaller than their siblings that had been in water for 2 days (lower PC1 scores; X2 = 27.58, p < 0.0001, Fig. 4B), but had wider tail muscle (higher PC3 scores: X2 = 9.31, p = 0.002, Fig. 4C); their PC2 scores were similar (X2 = 2.16, p = 0.14). Foam-nesting larvae had gut coils packed with yolk, while their unfed aquatic siblings had no visible yolk remaing; their gut coils were filled with brown material, with evidence of feces in the anal tube (Fig. S1A, S1C).

Figure 4 Short-term effects of extended terrestriality and foam-making on the morphology of Leptodactylus fragilis larvae after 12.5 d in the parental foam nest on soil.

(A) PCA biplot showing the first and third PC which encompass 78.9% of the variation in morphology among 14.5 d larvae after 2 d in water (blue centroid) or constructing a larval foam nest (black centroid). (B, C) Comparison of PC1 and PC3 scores between siblings that created one foam nest (black) or were transferred to water without making a foam nest (blue). (D) PCA biplot showing the first two PC which encompass 81.9% of the variation in morphology among larvae that made one (black centroid), two (brown centroid), and three (purple centroid) new foam nests. (E, F) Comparison of PC1 and PC2 scores among siblings across number of nests constructed. Contributions of original morphometric variables to each PC are displayed by arrows in biplots. Box plots show median, first and third quartiles, and extent of data to 1.5 X IQR; data points are also shown. Sample sizes (N) are indicated in biplot panels.

Comparing across terrestrial larvae that had made 1 to 3 foam nests (age 14.5–18.5 d), PC1, 2 and 3 accounted for 65.97%, 16.04% and 6.62% of variance in morphology, respectively, (Fig. 4D, Table S3). Seven measurements of body size contributed strongly to PC1, all with positive loadings, with total length loading most heavily (Fig. 4D, Table S4). Tail muscle height and tail length made the largest contributions to PC2, with positive loadings. Interorbital distance (positive loadings) and tail muscle width (negative loadings) contributed most to PC 3. Number of foam nests constructed did not affect scores on PC1 (X2 = 3.95, p = 0.14, Fig. 4E), PC2 (X2 = 1.07, p = 0.58, Fig. 4F) or PC3 (X2 = 1.39, p = 0.50). However, the appearance of gut coils indicates a reduction in yolk reserves with prolonged terrestriality and repeated nest construction, and no evidence of food passage through the gut (Figs. S1A, S1B).

Carry-over effects of extended terrestriality and foam-making on larval development to metamorphosis

Measured from oviposition, age at forelimb emergence was explained (R2m = 0.57; R2c = 0.71) by larval nest-construction (X2 = 105.61, p < 0.0001), but not by sibship size (X2 = 1.87, p = 0.17). It was marginally different for larvae that made zero or one foam nest (Fig. 5A; p = 0.04), but longer for those that made three nests (Fig. 5A; 0 vs. 3, p < 0.001; 1 vs 3, p < 0.001). The aquatic larval period (water entry to forelimb emergence) was between 15 and 28 days (Table S5). Variation in this period was not explained (R2m = 0.07; R2c = 0.36) by larval foam-making (X2 = 5.20, p = 0.07) or sibship size (X2 = 1.87, p = 0.17). Tadpoles that made one or three nests had an aquatic larval period similar to siblings that made zero nests (Fig. 5A; 0 vs 1, p = 0.20; 0 vs 3, p = 0.91; 1 vs 3, p = 0.07).

Figure 5 Carry-over effects of extended terrestriality and foam-making on time to and size at metamorphosis for Leptodactylus fragilis.

Terrestrial larvae entered the water after making zero (blue), one (black), or three (purple) new foam nests. Times from oviposition (age) and water entry (aquatic period) to forelimb emergence (A) and tail resorption (B). Box plots show median, first and third quartiles, and extent of data to 1.5 X IQR; data points are also shown. Total length at forelimb emergence in relation to age (C), aquatic period (E), and number of siblings in the nest (G). Snout–vent length at tail resorption in relation to age (D), aquatic larval + metamorphic period (F), and sibship size (H). Scatter plots show regression lines for each number of larval nests constructed. Sample sizes (N) are indicated for time and size data.

Measured from oviposition, age at tail resorption was explained (R2m = 0.47; R2c = 0.65) by larval nest construction (X2 = 64.22, p < 0.0001), but not by sibship size (X2 = 2.31, p = 0.12). At tail resorption, individuals that made one nest were similar to siblings that made zero nests (Fig. 5B; p = 0.88), but younger than those than made three nests (Fig. 5B; 0 vs 3, p < 0.001; 1 vs 3, p < 0.001). The period from water entry to tail resorption was 20 to 33 days (Table S5). Variation in this time was not explained (R2m = 0.04; R2c = 0.36) by larval nest-making (Fig. 5B; X2 = 0.99, p = 0.61) and only marginally by sibship size (X2 = 2.31, p = 0.05).

Total length of larvae at forelimb emergence was explained by a model that included a negative effect of the aquatic larval period (R2m = 0.12; R2c = 0.38; X2 =4.41, p = 0.04, Fig. 5E), but not age or larval foam-making (Fig. 5C). This model also included sibship size with a positive effect on total length (X2 = 4.19, p = 0.04; Fig. 5G). Snout–vent length (SVL) at tail resorption was best explained by a model that included a negative effect of either age or aquatic period (R2m = 0.15; R2c = 0.47; X2 = 6.68, p = 0.01; R2m = 0.12; R2c = 0.48; X2 = 3.92, p = 0.05, respectively; Figs. 5D, 5F), but neither model included an effect of larval foam-making. In these models, number of siblings in the group had only a marginal positive effect on SVL at tail resorption (age: X2 = 3.78, p = 0.05; aq. + metamorphic period: X2 = 3.36, p = 0.07; Fig. 5H), with no interaction effects.

Mass at forelimb emergence was best explained (R2m = 0.68; R2c = 0.74) by a model including a positive effect of total length at forelimb emergence (X2 = 91.62, p < 0.0001; Fig. 6A), a negative effect of either aquatic period (Fig. 6C) or age (X2 = 5.43, p = 0.02; X2 = 8.02, p = 0.005, respectively), and a marginal positive effect of sibship size (X2 = 3.05, p = 0.08, Fig. 6E). Mass at tail resorption was best explained by a model (R2m = 0.76; R2c = 0.83) including a positive effect of SVL at tail resorption (X2 = 94.30, p < 0.0001; Fig. 6B) and a marginal interaction with sibship size (X2 = 3.52, p = 0.06). The number of siblings in the foam, the time from water entry to tail resorption or age at tail resorption did not affect mass at tail resorption once SVL was included (Figs. 6D, 6F). Larval nest construction did not affect mass at forelimb emergence or tail resorption.

Figure 6 Carry-over effects of extended terrestriality and foam-making on mass at metamorphosis for Leptodactylus fragilis.

Terrestrial larvae entered the water after making zero (blue), one (black), or three (purple) new foam nests. Mass at forelimb emergence is plotted in relation to total length (A), aquatic period (C), and sibship size (E). Mass at tail resorption is plotted in relation to snout–vent length (B), aquatic larval + metamorphic period (D), and sibship size (F). Data points represent individuals, under each nest-making treatment, across sibship ID, and lines are regression fits for each nest number. Sample sizes (N) are indicated.

Discussion

Across ages, terrestrial larvae of L. fragilis are highly capable of constructing and maintaining their own foam nests and have sufficient energy reserves to construct multiple entirely new nests. While substantial research has examined parental strategies to construct developmental environments for offspring, with ecological and evolutionary consequences (Badyaev & Uller, 2009; Laland, Odling-Smee & Endler, 2017), our results highlight the importance of developmental environments constructed by early life stages themselves. Foam-making appears metabolically expensive, evidenced by very high levels of ammonia and, for older larvae, urea in new larval nests. Larvae grew more in water, compared to their siblings constructing foam nests on land whose size and morphology remained similar over an extended terrestrial period and construction of multiple nests, suggesting arrested growth and development. Because yolk reserves are the sole, limited energy source for non-feeding larvae (Womble, Pickett & Nascone-Yoder, 2016), we hypothesized that energetic costs of nest construction and urea excretion on land would carry-over to affect aquatic larval period or size at metamorphosis. Our results did not support this hypothesis, suggesting that energetic reserves in L. fragilis eggs are sufficient to meet these metabolic costs during developmental arrest over an extended period on land.

Foam-making ability and prolonged larval survival on land

Foam nests provide a critical, parentally constructed microhabitat that enables embryos, then larvae, of Leptodactylid frogs to survive for prolonged periods on land (Downie, 1984; Heyer, 1969; Méndez-Narváez, Flechas & Amézquita, 2015). However, over time and with drying, terrestrial nests that parents provide deteriorate, and they can dissolve in rain without freeing larvae into pools (Caldwell & Lopez, 1989; Downie, 1984). Our results highlight the value of larval foam-making, an ability documented in only a few foam-nesting frogs (Giaretta et al., 2011; De Almeida Marinho et al., 2022), in facilitating survival through a prolonged period on land. Leptodactylus fragilis larvae can construct an entirely new replacement nest within a day of hatching and retain this ability for at least 2 weeks longer. Contrary to our prediction, we found no evidence that larval foam-making ability increases with age, as the likelihood and extent of parental foam loss or deterioration increases. Based on foam volume produced, it appears similar from age 4.5–18.5 d, suggesting that larvae are fully competent to reconstruct the microhabitat they need to survive in soil through this entire period. Nonetheless, our experimental protocol may not have captured more subtle differences in the speed with which larvae replaced their nests. Whereas parental foam is formed by beating the oviductal secretion, larval foam in the L. fuscus group is created from the secretions of mucous glands in the buccal epithelium, which have been described at Gosner stage 25 (Downie, 1989; Giaretta et al., 2011). In L. fragilis the larvae clearly blow bubbles (Video S2), but body wriggling and tail movements might also contribute to new foam, as described for related species (Downie, 1989; Kokubum & Giaretta, 2005). Over time larval activity seems to decrease; once the foam has reached its maximum volume (~24 h) fewer larvae are visible at the surface, blowing bubbles (Videos S1, S2).

Although foam-making appears metabolically expensive, it may also be highly adaptive, particularly when incomplete flooding occurs and larvae must remain in the soil awaiting another rainfall. We do not know how long new larval nests can last, but considering their small size it seems likely that larvae may need to continually or repeatedly produce foam, either to maintain their nests or to replace them if another flooding event fails to release them from their chamber. This may explain why, across multiple nest-construction events, their ability did not change. Well-grown aquatic tadpoles from species with terrestrial foam nests survive longer on a damp surface out of water, compared to aquatic foam-nesters (Downie & Smith, 2003; Venturelli, Da Silva & Giaretta, 2021). However, these more developed tadpoles have likely already lost their foamy glands; foam-making ability declines during aquatic life (Downie & Smith, 2003) except in endotrophic Adenomera sp. tadpoles which maintain it until near metamorphosis (Kokubum & Giaretta, 2005). Those aquatic tadpoles still lose wet mass and suffer 35% mortality within 48 h on land (Downie & Smith, 2003; Venturelli, Da Silva & Giaretta, 2021). This constrasts strikingly with zero mortality of L. fragilis larvae removed from parental nests, even over 6 days of repeated larval nest construction.

The high variability of rainfall may have selected for a high foam-making capacity across early larval development and through a period of developmental arrest in L. fragilis. Even short periods without rain are associated with increased risk of mortality by dehydration for terrestrial embryos in three Neotropical frog lineages: Dendropsophus (Touchon & Warkentin, 2009), Centrolenidae (Delia, Ramírez-Bautista & Summers, 2013), and Phyllomedusinae (González, Warkentin & Güell, 2021; Salica, Vonesh & Warkentin, 2017). These lineages exhibit parental strategies, including oviposition site plasticity and parental care, to prevent embryo mortality by dehydration (Delia et al., 2019; Delia, Bravo-Valencia & Warkentin, 2020; Touchon, Urbina & Warkentin, 2011; Touchon & Warkentin, 2008; Touchon et al., 2024). However, their embryos can also hatch and enter the water earlier when faced with dry conditions (Salica, Vonesh & Warkentin, 2017; Touchon & Warkentin, 2010) high temperature (Guevara-Molina, Gomes & Warkentin, 2022) or high ammonia levels (Lisondro-Arosemena, Salazar-Nicholls & Warkentin, 2024), and only extend their development on land under good hydration (Delia et al., 2019). In other amphibians and fishes with terrestrial eggs, developmental arrest and low embryonic metabolism have been associated with prolonged time on land while waiting for a flooding cue to hatch (Bradford & Seymour, 1985; Martin, 1999; Petranka & Petranka, 1981). In such species, metabolic costs of soil dehydration have been reported, with effects on embryonic growth and development (Seymour, Geiser & Bradford, 1991), but costs were not assessed at later stages. The foam-making ability of L. fragilis larvae functions as an adaptive behavioral response to variability in their terrestrial environmental and the loss of extended benefits conferred by a parentally provided structure.

The ecological and physiological importance of self-constructed larval microenvironments are recognized in insects (e.g., Tonelli et al., 2018; Baer & Marquis, 2020) but less studied in vertebrates. We are not aware of studies addressing the ability of early larvae to modify their developmental environment in other frog lineages, or their potential consequences during development. Indeed, larval foam-making has been previously described in just a few groups of leptodactylids, all with initial or complete larval development in subterranean foam nests (Caldwell & Lopez, 1989; Downie, 1984; Giaretta et al., 2011; Kokubum & Giaretta, 2005). It would be worth assessing the selective factors and fitness consequences of foam-making from the perspective of both parents and offspring in terrestrial-breeding leptodactylids, in particular within an extended phenotype and niche construction framework, given this apparent parent-offspring convergence in ecology and behavior (Badyaev & Uller, 2009). Both parental and larval foam nests may facilitate niche exploitation, by increasing fitness (both stages) and survival (offspring), and have facilitated aquatic to terrestrial breeding transitions in some members of the family Leptodactylidae (Heyer, 1969; Méndez-Narváez, Flechas & Amézquita, 2015; Méndez-Narváez & Warkentin, 2022). For instance, foam may slow water loss, reducing dehydration risk (Zina, 2006), provide a thermal buffer (Méndez-Narváez, Flechas & Amézquita, 2015), and air bubbles in foam may facilitate oxygen uptake to sustain metabolism in densely packed groups (Philibosian et al., 1974; Seymour & Loveridge, 1994; Seymour & Roberts, 1991). Currently, the increasing frequency of short periods without rainfall during the rainy season (Touchon & Warkentin, 2009) is likely increasing both foam nest dehydration and the need for larval foam-making, increasing the metabolic costs associated with preventing ammonia toxicity (Candelas & Gomez, 1963; Méndez-Narváez & Warkentin, 2022, 2023; Shoemaker & McClanahan, 1973).

Nitrogen excretion in larval foam nests

Terrestrial embryos and larvae can face a waste-disposal problem, as nest dehydration increases ammonia concentration in their developmental environments, increasing the risk of toxicity (Méndez-Narváez & Warkentin, 2022). At 12.5 d, nest-dwelling larvae of L. fragilis increase urea excretion under dry conditions in soil (Méndez-Narváez & Warkentin, 2022) and their tissues exhibit high activity levels of two key urea cycle enzymes, CPSase 1 and arginase (Méndez-Narváez & Warkentin, 2023). Here we found that, over just 2 days, very high concentrations of ammonia and urea accumulated in new larval foam nests, and arginase activity was high in larval tissues. For comparison, at 12.5 d parental nests in dry soil contained 53.5 ± 48.9 mmol/L ammonia and 59.2 ± 71.0 mmol/L urea-N (Méndez-Narváez & Warkentin, 2022). Here, larvae from such conditions produced nests containing 195.8 ± 150.1 mmol/L ammonia and 461.6 ± 510.4 mmol/L urea-N, over 3-fold and 8-fold higher, respectively. Although larval nests are smaller, these high concentrations are largely due to greater amounts of nitrogen waste accumulated per larva (ammonia: 0.01 ± 0.01 vs. 0.03 ± 0.02 μmol; urea: 0.01 ± 0.01 vs. 0.08 ± 0.07 μmol over 12.5 d in parental vs. 2 d in larval nests, respectively; Méndez-Narváez & Warkentin, 2022; this study). Considering that these non-feeding larvae have arrested growth and development, which reduces metabolism, their very rapid production of N-wastes indicates rapid catabolism of yolk proteins to support other functions—presumably both the activity required for foam production to build a new nest and the cost of urea synthesis to avoid ammonia toxicity in it.

The higher metabolic cost of urea synthesis through the urea cycle, compared to ammonia excretion (Shambaugh, 1977), may favor plasticity in N-waste excretion with time on land, moisture availability, and foam-making efforts, to reduce this cost when possible. Here, we did not find significantly more ammonia or urea in nests produced by larvae from dry soils; rather, their higher concentrations were correlated with lower water volumes. Nonetheless, urea excretion was explained by predicted ammonia levels in new larval nests, as it was in parental nests (Méndez-Narváez & Warkentin, 2022). Dry conditions on land may favor earlier and/or greater urea excretion, as ammonia concentrates in nests losing water and the metabolic cost of larval foam-making, to replace deteriorating parental foam, is met by protein catabolism that increases N-wastes (Méndez-Narváez & Warkentin, 2022). A similar scenario may occur if early flooding fails to release larvae into ponds, so that wastes produced as trapped, nestless larvae make new foam accumulate in a small volume of foam. Thus, larval foam-making may be a key trait that both enables survival on land and necessitates a high capacity for ammonia detoxification. However, we found no urea, only ammonia, accumulated in foam nests produced by 4.5 d larvae, suggesting these early larvae lack the enzymatic mechanisms to synthesize urea. This is consistent with our previous findings of no urea in parental nests at age 8.5 d (Méndez-Narváez & Warkentin, 2022). Nonetheless, ammonia levels in nests produced by 4.5 d larvae were over twice as high as levels accumulated in parental nests after 12.5 d in dry soil, and half of them were well into the concentration range where aquatic L. fragilis larvae die (Méndez-Narváez & Warkentin, 2022), again emphasizing the metabolic cost of foam-making. These ammonia levels did not increase further, for older larvae (12.5 d) or with sequential nest construction; rather, larvae began to excrete and accumulate urea. Even so, the ammonia levels that accumulated over 2 days, as 12.5 d larvae constructed foam nests, were often within the range that caused 50% mortality of 12.5 d larvae in aquatic ammonia solutions (95% CI for 48 h-LC50: 108–122 mmol/L; Méndez-Narváez & Warkentin, 2022). Despite this, we observed no larval mortality during our nest-construction trials. This suggests the ammonia tolerance of L. fragilis terrestrial larvae substantially exceeds that reported for other anuran larvae, including even the high tolerance found for age-matched aquatic L. fragilis (Méndez-Narváez & Warkentin, 2022).

As with N-waste excretion, we found arginase activity in larval tissues to be higher for 14.5 d larvae that made new nests compared to earlier measurements for 12.5 d larvae in parental nests (on wet soil: 0.08 vs. 0.07, on dry soil: 0.11 vs. 0.06 μmol min−1 mg−1, respectively; this study and Méndez-Narváez & Warkentin, 2023), suggesting a higher metabolic cost of urea synthesis by the urea cycle. This has been also suggested in tadpoles that experience pond drying and can survive on humid soil for several weeks (Candelas & Gomez, 1963; Venturelli, Da Silva & Giaretta, 2021; Shoemaker & McClanahan, 1973). The urea cycle may also occur in other contexts with high risk of ammonia toxicity. For instance, we found arginase activity in nest-makers from dry soil close to that in L. fragilis larvae after 4 days in high-ammonia water (Méndez-Narváez & Warkentin, 2023: 0.13 μmol min−1 mg−1, measured at 12.5 d). Although arginase, which produces urea from arginine, catalyzes the last step in the urea cycle (Brown, Brown & Cohen, 1959), this enzyme has other metabolic roles (Srivastava & Ratha, 2010). Energetic costs of both foam-making activity and ammonia detoxification would presumably accumulate as larvae remain on land longer, drawing on their yolk reserves. It would be worth studying other physiological consequences of high activity during foam-nest construction or exposure to high ammonia that may affect long-term survival, such as oxidative stress and antioxidant activity, where the relationship with the energy budget is complex (Burraco, Díaz-Paniagua & Gomez-Mestre, 2017; Guo et al., 2023; Zamora-Camacho et al., 2023).

Short-term consequences of extended terrestriality and foam-making

We found that prolonging larval life on land has immediate effects on growth and morphology; larvae were larger after 2 days in water compared with age-matched siblings that remained on land making a new foam nest, but they had relatively less muscular tails and had depleted their yolk reserves. Environmentally cued early hatching and transition to water has immediate consequences for the growth and development of arboreal frog embryos, with earlier feeding and faster growth of aquatic larvae compared to later-hatching embryos (Delia et al., 2019; Warkentin, 1999). The size difference we found between age-matched aquatic and terrestrial larvae resembles that between the aquatic larvae of the grunion fish, Leuresthes tenuis, vs. their terrestrial embryos, which largely arrest development to prolong survival while awaiting flooding (Moravek & Martin, 2011). Closely related Leptodactylus latinasus and L. bufonius larvae in foam nests also showed slower yolk utilization and grew less than their aquatic larvae (Philibosian et al., 1974). Arboreal egg clutches may impose physiological constraints on metabolism, as suggested to explain the accelerating effect of hatching on growth and differentiation in pre-feeding A. callidryas (Warkentin, 1999, 2007). Terrestrial foam nests might also impose such constraints, including metabolic depression due to accumulation of urea in tissues (Muir, Costanzo & Lee, 2008). Large yolk reserves are hypothesized to extend embryonic survival times on land in L. tenuis and the frog Pseudoprhyne bibronii (Bradford & Seymour, 1985; Moravek & Martin, 2011). Terrestrial leptodactylid embryos also have large yolk reserves, compared to aquatic-nesting frogs in the same family (Méndez-Narváez, 2012; Pereira et al., 2015). Rapid consumption of these reserves could explain the rapid growth observed for initially unfed L. fragilis larvae in water, as in A. callidryas, L. tenuis, and other species of the L. fuscus group (Moravek & Martin, 2011; Warkentin, 1999; Philibosian et al., 1974). Moreover, even in apparently clean water with no food added, it appears that L. fragilis larvae may ingest something, which could also contribute to growth.

Extended development in arboreal eggs has been associated with morphological changes that benefit larvae when they transition to water, such as larger, stronger tails that improve swimming and gut development that reduces time to external feeding (Delia et al., 2019; Warkentin, 1999). Previous studies have also found higher mortality of smaller, less developed larvae when exposed to aquatic predators (Gomez-Mestre, Wiens & Warkentin, 2008; Vonesh, 2005; Warkentin, 1995), although larval mortality depends on the types of predators in a pond, which may differentially consume different prey sizes (Willink et al., 2014). In contrast, we found no increase in larval size or change in form with prolonged time on land, comparing 14.5 to 18.5 d siblings that made one to three nests, suggesting this delay confers no benefit upon entry into water. However, our design cannot separate effects of time on land from those of foam-making activity per se, thus the effects of prolonged terrestriality in a persistent parental nest may differ. There is evidence that continued terrestrial development during the first four days after hatching at about 3.5 days may benefit larvae. During this period, nest-dwelling larvae of L. fragilis increase in size and show substantial morphological change; then their growth and development seem to stop (N Belduque-Correa, KM Warkentin & J Méndez-Narváez, 2019, personal observations). Their terrestrial persistence after 7 days could be considered a period of developmental arrest, as described for nest-dwelling larvae of L. fuscus (Downie, 1994), although older larvae of L. fragilis are clearly still behaviorally and metabolically active, as evidenced by their foam-making capacity and excretion of N-wastes (Méndez-Narváez & Warkentin, 2022, 2023). Grunion embryos show a linear increase in metabolism during development to hatching competence, then can remain in a steady high metabolic state for several weeks, using oil from yolk reserves (Darken, Martin & Fisher, 1998). In contrast, a 90% decrease in oxygen consumption can occur with developmental arrest of somite proliferation and DNA content in the annual killifish Austrofundulus limnaeus (Podrabsky & Hand, 1999). As suggested for L. tenuis embryos, it may be advantageous for L. fragilis larvae to retain metabolic activity in an unpredictable terrestrial environment, both to make new foam if needed and to take advantage of brief flooding opportunities to escape from their chambers.

Effects of extended terrestriality and foam-making on development to metamorphosis

Size advantages evident when larvae enter the water do not necessarily persist; compensatory growth can occur during aquatic development of initially smaller larvae, allowing them catch up, either rapidly or at some point before metamorphosis (Touchon et al., 2013; Touchon & Warkentin, 2010). On the contrary, reduced food availability and high larval density can both delay metamorphosis and result in emergence of smaller, shorter-legged frogs (Bouchard et al., 2016; Gomez-Mestre et al., 2010). We found no evidence for compensatory growth in larvae that spent two or six extra days (from age 12.5 to 14.5 or 18.5 d) on land, making new foam nests, nor did these conditions affect morphology at metamorphosis. Neither did we find any long-term delay. These larvae had a similar aquatic larval period and metamorph size and weight as their siblings that made no new nest and entered the water at age 12.5 days. Our results suggest that terrestrial L. fragilis larvae can cope with a prolonged period on land and associated metabolic costs without long-term consequences for fitness-relevant phenotypic traits and development to metamorphosis. They seem to avoid potential carry-over effects on growth and development as well as physiological consequences, documented in other species, that can reduce survival (Burraco, Díaz-Paniagua & Gomez-Mestre, 2017).

Larval adaptations and parental investment in egg size may mitigate costs of extended terrestrial development on subsequent aquatic development to metamorphosis in L. fragilis. These terrestrial endotrophic larvae can use their large yolk reserves for energy to construct new foam nests and synthesize urea to prevent ammonia toxicity. By arresting development after 7 d, larvae enter water at a similar size and shape even after building multiple foam nests; alternatively, if young larvae enter the water they experience an initial period of rapid growth, which may also be enabled by their large yolk reserves. However, as we did not test the youngest hatchling larvae, with the largest yolk reserves at their earliest possible time of water entry, we do not know if our observed metamorphic phenotypes are already carrying an unmeasured cost of prolonged terrestrial life or, alternatively, may represent a form of canalized phenotype.

We found variation in metamorph size that was not explained by larval foam-making but was correlated with aquatic period and, in some cases, sibship size. In some sibships larvae grow fast and metamorphose early and large, while in others they grow slower and metamorphose later and smaller. In anurans, larger body size at metamorphosis is associated with higher fitness in post-metamorphic stages (Cabrera-Guzmán et al., 2013; Gomez-Mestre et al., 2010; Scott et al., 2007). However, a negative correlation of size with age, as in our study, can occur when larvae experience growth-constraining conditions before the onset of feeding (Bouchard et al., 2016; Gomez-Mestre et al., 2010). It is worth assessing benefits of both larger size and younger age at metamorphosis, as slower-growing larvae may sacrifice size to limit their larval period. Moreover, exposing aquatic larvae to harsh environmental conditions (e.g., pond drying) can trigger plastic changes in size and time to metamorphosis (Gomez-Mestre et al., 2010), which would not be evident under our experimental conditions. As L. fragilis larvae develop in temporary ponds with high risk of drying, short larval periods may sometimes be crucial for survival, as in other frogs that must either arrive early or metamorphose quickly to escape pond drying (Laurila & Kujasalo, 1999; Murillo-Rincón et al., 2017). This contrasts with species that use longer-lasting pools and, without such time constraints, may have longer and more broadly variable larval periods (Bouchard et al., 2015; Touchon et al., 2013).

Our results suggest that the initial clutch size, a maternal effect, may have consequences for nest-building larvae and at metamorphosis. Evolutionary and intraspecific changes in clutch characteristics, such as the number and size of eggs, have been studied in the context of allocation of reproductive effort, parent-offspring co-evolution, and evolutionary transitions in reproductive modes and parental care behavior (Delia, Bravo-Valencia & Warkentin, 2020; Gomez-Mestre, Pyron & Wiens, 2012; Kasimatis & Riginos, 2016). We found that larvae in larger families produced less foam per capita, suggesting the possibility of energy-saving benefits of collective action. This might explain the trend toward a positive effect of sibship size on size at metamorphosis. This is opposite to more commonly studied group size or density effects, such as those mediated by intraspecific competition for food in aquatic environments. For instance, high larval density or low per capita food availability can affect larval nutritional traits (fat bodies and gut morphology) with lasting effects on metamorph or juvenile body size (Bouchard et al., 2015; Gomez-Mestre et al., 2010). The shared construction and maintenance of their essential microhabitat by endotrophic larvae may fundamentally alter the impacts of group size.

Conclusions

When life stage transitions are constrained by unpredictable external factors, embryos and larvae may evolve active coping strategies. We hypothesized that plastic larval responses that enable extended terrestrial survival in L. fragilis have energetic costs that carry over to affect the aquatic larval period and size at metamorphosis. We found that nest-dwelling larvae of L. fragilis are well-adapted to respond to terrestrial challenges, including dehydration risk, accumulation of N-wastes, and loss of their original parental foam nest. These unfed larvae rely on their yolk reserves while awaiting flooding of their nest chambers. Although they enter developmental arrest, ceasing morphological change, larvae are metabolically and behaviorally active. Sibling groups can quickly construct a new foam nest, multiple times if needed, starting soon after hatching, with no evidence for changes in their ability to do so over a 2-week period. Moreover older larvae show a high capacity for urea synthesis that functions to prevent ammonia toxicity. Compared to siblings on land making foam nests, aquatic larvae rapidly increase in size, even without food provided, probably by a shift in yolk utilization. However, carry-over effects of extended terrestriality and nest-making were not apparent at metamorphosis. Large yolk reserves and temporary developmental arrest seem to allow unfed larvae to meet high energy demands during active periods on land and may explain this apparent lack of lasting carry-over effects on development. Nonetheless, we found some effects on metamorph size that could be mediated by maternal clutch-size effects in cooperative groups, when siblings face challenges together during early development on land.

Our work emphasizes the value of studying changes in metabolism and behavior, in addition to morphological traits not captured by standard staging tables (Moravek & Martin, 2011; Warkentin, 1999), in anamniotic vertebrates that have evolved plastic extensions of terrestriality in early life. Our results also suggest that terrestrial early life stages can construct and modify their microenvironment and alter their physiology to survive harsh conditions, or take advantage of more benign ones, without necessarily paying long-term costs. These performance traits of early life stages could be just as important as parental strategies for enabling evolutionary colonization of new environments. We suggest that further study of the behavioral and physiological ecology of early stages of semi-terrestrial vertebrates across habitat transitions would be valuable. As well as performance traits facilitating terrestrial survival, further studies could assess costs and potential carry-over effects in the context of subsequent challenges for aquatic larvae, such as predation risk, nutritional challenges, and pond drying. Elucidating the adaptive plastic responses of early life stages to their variable environments across terrestrial-to-aquatic transitions, and the context-dependent longer-term fitness consequences of those responses, would improve our understanding of the evolutionary importance of changes in early life stages for the reproductive colonization of land. It would also be useful for predicting responses and understanding vulnerability in the context of climate and habitat change that are altering developmental environments for amphibians.

Supplemental Information

Supplemental Information 1 Sibling larvae of Leptodactylus fragilis after 12.5 d in the parental foam nest on soil, then larval nest construction or water entry.

Larvae at (A) 14.5 days, after constructing a new larval foam nest, (B) 18.5 days, after constructing three larval nests. and (C) 2 days in aged, dechlorinated tap water with no food provided.

Supplemental Information 2 Time lapse recording of a sibling group of terrestriallarvae of Leptodactylus fragilis making a new foam nest in a Petri dish.

Larvae were removed from their parental nest at age 12.5 days and placed together in a vertical Petri dish without water. They were photographed every minute for 15 hours with a Canon EOS 70D and 100 mm macro lens (f14, 1/100 s); the video represents 45 min per sec. In experiments, we assessed foam-nesting performance after 48 h.

Supplemental Information 3 Larval foam-making behavior in Leptodactylus fragilis.

Larvae are 12.5 days old, still in their parental nest in dry soil. The camera is focused on the center of the nest, where larvae are constructing new foam to maintain their deteriorating parental nest. During one minute of recording, four larvae are visible actively blowing bubbles at the surface of their foam nest, while another larva remains motionless. Rapid, variable heartbeats are evident in several bubble-blowing larvae. The video plays in real time.

Supplemental Information 4 Supplementary Tables.

Supplemental Information 5 Raw data.

Data that were used to test larval foam making ability, nitrogen excretion in larval nest, arginase activity of larval tissues in larval nests, short term effect on larval size and carry over effect on larval development and metamorphosis.

We thank Carolina Amorocho for field assistance, Team Treefrog for sharing experiences during field work and the Gamboa Frog Group and allies for their intellectual support in Panama. We thank Jack Friend for his assistance measuring larval morphology. We also thank Rachel Page and Roberto Ibañez for enabling this research in Panamá. Nitrogen quantifications were conducted in the Biogeochemistry lab in Panama, which was led at the time by Dr. Ben Turner. We also thank Richard Wilkie and Oana Birceanu for training JMN in enzymatic analysis at Wilfrid Laurier University and Jennifer Bhatnagar and Corrine Vietorisz for facilitating laboratory work at Boston University. We thank Javier’s PhD committee—Chris Schneider, Jennifer Bhatnagar, Sean Mullen, Ivan Gomez-Mestre, and Peter Buston—for their intellectual support. We thank Dr. Diego Venturelli and two anonymous reviewers for their comments that helped to improve this manuscript.

Additional Information and Declarations

Competing Interests

The authors declare that they have no competing interests.

Author Contributions

Javier Méndez-Narváez conceived and designed the experiments, performed the experiments, analyzed the data, prepared figures and/or tables, authored or reviewed drafts of the article, and approved the final draft.

Karen M. Warkentin conceived and designed the experiments, authored or reviewed drafts of the article, analyzed the data, prepared figures and/or tables, and approved the final draft.

Animal Ethics

The following information was supplied relating to ethical approvals (i.e., approving body and any reference numbers):

Animal research was approved by the Smithsonian Tropical Research Institute, ACUC protocol # 2016-0520-2019A1–A3.

Field Study Permissions

The following information was supplied relating to field study approvals (i.e., approving body and any reference numbers):

Field research and experiments were approved by the Panamanian Ministry of the Environment, MiAmbiente permits SE/A-56-17, SC/A-51-18.

Data Availability

The following information was supplied regarding data availability:

The raw data are available in the Supplemental Files.

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
