# Peer review of "Effects of larval foam-making and prolonged terrestriality on morphology, nitrogen excretion and development to metamorphosis in a Leptodactylid frog"

_PeerJ, doi:10.7717/peerj.18990_

## Round 0.1 · original submission · Major Revisions

Dear authors, I ask you to respond very carefully to all the comments of the reviewers who have done a very deep analysis of your manuscript. I believe that this article can be published after all the shortcomings are eliminated.

Reviewer 1 ·

Basic reporting

This manuscript explains how the foam-making behaviour and terrestriallity affect morphology, development and metabolism of Leptodactylid frogs, which is well stated in the title. The background for this research is explained in detail in the Introduction with relevant references. Hypotheses are well-defined. The authors did a series of experiments and collected a lot of valuable data which was carefully examined and results are well presented. I have only minor suggestions considering these part of the manuscript. A lot of the important results are obtained and should be discussed, but the Discussion section is too long and hard to follow. I would suggest rewriting discussion to more summarize results and focus on the hypothesis given at the end of the Introduction.

Experimental design

The experimental design, obtained data and analyzes used for testing the hypotheses are very well presented and explained in detail.

Validity of the findings

Results are well presented, carefully examined for conclusions and backed up and connected with literature,

Additional comments

Some minor comments:

Lines 154-171: Add references to the first part of the sentences which refer to back up your hypotheses.
For example: “ (1) Because foam-making depends on the development and function of foamy glands and performance of foam- making behavior (REF), and because natural selection for foam-making may intensify as parental foam deteriorates over time (REF), we hypothesized that foam-making ability increases developmentally.”
Line 192: Is “d” for days? Does the reference of unpublished data refer to “before, at, and after the onset of developmental arrest”?
Line 199: It seems that something is missing in the sentence, maybe “to move TO a larger pond”?
Line 206: Suggestion: “as” instead of “;”
Line 207: Add a reference for “larvae can naturally spend prolonged periods in developmental arrest in their new foam.”
Lines 215-217: References?
Line 250: frozen
Line 349: Please provide R code.
Lines 479-481: References

·

Basic reporting

The text is well-written in clear, professional English, with accessible language suitable for a scientific audience. However, adjustments could be made to enhance readability for the reader. These points are highlighted below and in the PDF. The introduction provides appropriate context on terrestrial larval development in amphibians, with well-selected and relevant references that offer a solid foundation for the study. Some adjustments have been suggested to improve text construction and some references were suggested. The text structure follows PeerJ standards, with well-organized sections that contribute to clarity. The figures are high quality, well-labeled, and described, illustrating the main findings, although they are information-dense and could be simplified. Raw data has been made available, allowing for replication and verification of the obtained results.

Experimental design

The study is an original, primary research project within the scope of the journal. The research question is well-defined, relevant, and significant, addressing an identified gap in the literature by investigating the ability of L. fragilis larvae to build and maintain their own foam nests. The investigation was conducted with high technical and ethical rigor, adhering to approved ethical guidelines from the institution's ethics committee, as noted in the text. Methods are described with sufficient detail to allow replication of the study, but minor adjustments could improve the readability and clarity of each process for the reader.

Validity of the findings

The study makes a valuable contribution to scientific knowledge on terrestrial larval development in anurans, offering ecological and evolutionary insights. It also encourages further research and interdisciplinary collaborations with potential applications across other scientific fields. All underlying data are provided, robust and statistically sound. The conclusions are clearly stated, directly linked to the original research question.

Additional comments

The study examines the ability of Leptodactylus fragilis larvae to construct and maintain foam nests and explores its morphological, metabolic, and developmental implications. Results indicate that the larvae can renew the foam nest, though this activity is metabolically costly, leading to elevated ammonia and subsequently urea levels in the nests. Larvae that accessed water grew more than those that remained on land, although both groups showed similar size and morphology.
Overall, the work is well-written, with a complex experimental design and a robust statistical analysis, supporting its suitability for publication in PeerJ. However, certain points should be addressed to enhance clarity for the reader, as PeerJ covers a broad audience across various fields of knowledge.
I made many suggestions to enhance the text's fluidity, which are noted here (below) and in the PDF. Additionally, I would like to highlight some sensitive points in the manuscript that need revision, along with reiterating several questions and suggestions below.

Introduction
Line 125: A paragraph in the introduction contextualizing how foam nests are formed, and their phylogenetic context would be beneficial to improved introduction, even to justify the use of the species L. fragilis in the study. L. fragilis belongs to the L. fuscus group, in which all species exhibit foam-nesting behavior. Additionally, L. fuscus tadpoles can also produce new foam and survive in aggregated groups for several days out of water.

Line 61: The citation Wilbur & Collins, 1973, is missing from the reference list.

Line 138 and 145: There was a break in the sequence of figures, with references to Fig. 1A followed by 1C. Fig. 1B was not referenced in the text.

Line 154 – 171: The construction of the hypotheses and objectives is interesting; however, to improve the text’s clarity, the objectives should be mentioned before each hypothesis. For example: “The objectives of this study were: (1) to assess the foam-making ability of larvae across different developmental stages, with the hypothesis that this ability improves as larvae mature. This improvement may be linked to the progressive development and function of foamy glands, the refinement of foam-making behavior, and increased selective pressure to maintain foam as parental nest structures deteriorate over time. (2) …".

Materials & Methods
Section: Study site and experimental conditions
Line 178 – 180: More details are needed on how the foam nests were located and collected, as well as the methods used for their transport to the laboratory.

Line 182: Additional information about the maintenance environment is necessary, such as the size of the containers, the number of containers used, and whether there was one foam nest per container.

Section: Larval foam-making ability across ages.
This section lacks some details that would help clarify the division of experimental groups and, especially, the sampling process.
• Were all larvae from each foam nest used?
• Why was the number of larvae not standardized in each sample?
• Did you classify the larval stages? Although the group of tadpoles is the same age, some develop faster or slower than others. Was this observed?

Section: Nitrogen excretion in larval nests during prolonged terrestriality.
The text begins by mentioning that a different set was used from the previous section but does not specify how the foam nests were maintained, particularly to achieve the two experimental conditions of moist and dry soil. The age divisions and sample size were also not justified. Why weren’t nests in dry soil tested at 4.5 days?
Since the study involves a dense experimental section with many procedures, it would be beneficial to improve this part of the text to facilitate the reader's understanding.

Line 245: Was tissue extracted from all the larvae? If not, from how many larvae was the tissue extracted? Was there randomization in the selection of these larvae? Did these larvae build their nest inside the parental nest, or did they come from the parental nests under dry and wet conditions? This terminology of parental foam nest is presented throughout the text, and it confuses understanding. Perhaps I can replace it with another definition, for example, original nests and renewed nests

Section: Carry-over effects of prolonged terrestriality and larval foam-making.
Since this section uses the larvae from the foam nests in the "Larval foam-making ability across ages" section, I believe it should come before the "Nitrogen excretion in larval nests during prolonged terrestriality" section. I think this will help clarify which set of egg clutches the experiment was conducted on.

Line 279: What was the range of developmental stages obtained in each condition? This is an interesting data point to present because, although the animals are the same age, they may exhibit different stages as some tadpoles develop faster than others. This developmental advancement in some tadpoles might explain the large variation or outliers in your data. If you have this information, it would be helpful to include it.

Line 285: Was it possible to measure the yolk sac, even after 12 days? It usually disappears after stage 22 (Gosner, 1960), doesn’t it? In the video from the supplemental material, it is already possible to see the tadpoles with their intestines formed and their gills internalized.

Line 291 and 353: There was a break in the sequence of figures, with references to Figure 1D followed by Fig.2. Fig.1E was not referenced in the text.

Line 303: “Froglets were kept …” This sentence, in its current position, created confusion in interpreting the following sentences. Moving it to the end of the section would make the text clearer.

Section: Statistical analysis
The statistical approach used in this study is extensive and well-founded, involving linear mixed effects models (LMEMs), generalized linear mixed models (GLMMs), combining multivariate analyses (PCA), mixed effects linear models (LMEMs), and methods for model selection and assumption validation. These are well designed to address the study's questions. The suggested adjustments below are merely incremental improvements to enhance the transparency and reliability of the results.
My suggestions are:
• To increase the reliability of the LMEMs, checking the normality and homogeneity of the residuals of the fitted models would be important. If the residuals show significant deviation from normality, data transformations or alternative models (such as GLMMs) may be considered.
• Given that multiple variables and covariates are involved, adding interactions between these variables in the models could reveal more complex dependencies among the predictor variables.
For reflection:
• The use of PCA to summarize morphological variation is appropriate for multidimensional data, as it allows for the identification of the axes of greatest variation among the larvae. However, only three traits stood out (two related to the tail and one related to the yolk), which would be the predictable traits. Perhaps an analysis using only these characteristics would reveal a more tangible biological meaning.
• LMEMs are used to compare principal component scores (PC1 and PC2) and test the effects of factors such as "foam" and "age" on morphological traits. Including "sibship" as a random factor is important for these tests; however, they may limit the author's autonomy in seeking variables that provide greater biological explanations of the topic, as they involve testing everything against everything, sometimes making comparisons that lack biological sense.

Results
Section: Larval foam-making ability
Interesting results: the more tadpoles there are, the greater the volume of foam; however, smaller groups of tadpoles produce more foam. Could this be because foam is produced by the friction between individuals? The more tadpoles there are, the more friction, the more foam. However, having too many tadpoles might also limit the volume of that foam, preventing it from expanding. On the other hand, with fewer tadpoles, their interference in the foam's expansion is less. Does this make sense to you? An interesting aspect to evaluate is the behavior of foam production; this has already been documented in adults, who need to "beat" the foam, but little is known about tadpoles.

Nitrogen excretion in larval foam nests
There is a very large variability in the data for ammonia and urea, especially for the first nest constructed.

Line 368: Although the test showed a significant difference, looking at the graph, this difference may have occurred because some nests had very low ammonia values, pulling the average down. Could this be related to the number of tadpoles per nest? The fewer the animals, the lower the levels of ammonia and urea? Additionally, the sample size for the third nest is very low. I believe it's worth highlighting this in the text and showing the limitations of the claims.

Section: Short-term effects on larval size and morphology
I reiterate my comment about exploring the actual morphological data, especially regarding the musculature of the tail and the yolk sac. Again, even after about 12 days, did the tadpoles still have yolk sacs?

Section: Carry-over effects of extended terrestriality and foam-making on larval development to metamorphosis
Here, several comparisons are made; which of them makes biological sense?

Discussion
Line 456 – 465: Present a summary of the results; there is no discussion here

Line 477: Although it is not widely studied, there is important research on this subject that need to be considered in your citations. Here is a list below for your review.
ALCOCER, L., X. SANTA CRUZ, H. STEINBEISSER, K. H. THIERAUCH, AND E. M. PINO. 1992. Ureotelism as the prevailing mode of nitrogen excretion in larvae of the marsupial frog Gastrothecario bambae (Anura, Hylidae). Comparative Biochemistry and Physiology 101:229–231.
BLACK, J. H. 1974. Larval spade foot survival. Journal of Herpetology 8:371–373
CANDELAS, G. C., AND M. GOMEZ. 1963. Nitrogen excretion in tadpoles of Leptodactylus albilabris and Rana catesbeiana. American Society of Zoologists 203:521–522.
CANDELAS, G. C., E. ORTIZ, C. VASQUEZ, AND L. FELLIX. 1961. Respiratory metabolism in tadpoles of Leptodactylus albilabris. American Society of Zoologists 1:348.
CRUMP, M. L. 1989. Effect of habitat drying on developmental time and size at metamorphosis in Hyla pseudopuma. Copeia 1989:794–797
DOWNIE, J. R., AND J. SMITH. 2003. Survival of larval Leptodactylus fuscus (Anura: Leptodactylidae) out of water: developmental differences and interspecific comparisons. Journal of Herpetology 37:107–115
GRAFE, T. U., S. K. KAMINSKY, AND E. LINSENMAIR. 2005. Terrestrial larval development and nitrogen excretion in the Afro-tropical pig-nosed frog, Hemisus marmoratus. Journal of Tropical Ecology 21:219–222.
ROCHA, C. F. D., D. C. PASSOS, AND W. E. MAGNUSSON. 2017. Survival of Amazonian tadpoles under harsh water-limiting conditions. South American Journal of Herpetology 12:212–217.
VALERIO, C. E. 1971. Ability of some tropical tadpoles to survive without water. Copeia 1971:364–365.
VENTURELLI, D. P., AND W. KLEIN. 2019. Effect of hydric stress on locomotion and morphology of tadpoles from temporary ponds. Journal of Experimental Zoology Part A 331:175–184
VENTURELLI, D.P, Da SILVA, W.R., GIARETTA, A.A. 2021. Tadpoles’ Resistance to Desiccation in Species of Leptodactylus (Anura, Leptodactylidae). Journal of Herpetology, Vol. 55, No. 3, 265–270, 2021

Line 491: Interesting! The literature shows the opposite effect: the higher the density, the faster and smaller the tadpoles develop.
NEWMAN, R. 1998. Ecological constraints on amphibian metamorphosis: interactions of temperature and larval density with responses to changing food level. Oecologia (1998) 115:9±16.
RICHTER-BOIX, A. TEJEDO, R, REZENDE, E. 2011. Evolution and plasticity of anuran larval development in response to desiccation. A comparative analysis. Ecology and Evolution.
WILBUR, H.M. 1977. Density-Dependent Aspects of Growth and Metamorphosis in Bufo americanus. Ecology

I believe the first part of the discussion could be improved, as it presents a summary of the results and arguments that the topic is under-researched. I suggest synthesizing or even removing it from the text.

Section: Foam-making ability and prolonged larval survival on land
Line 497: Review other references.
BLACK, J. H. 1974. Larval spade foot survival. Journal of Herpetology 8:371–373
CRUMP, M. L. 1989. Effect of habitat drying on developmental time and size at metamorphosis in Hyla pseudopuma. Copeia 1989:794–797
DOWNIE, J. R., AND J. SMITH. 2003. Survival of larval Leptodactylus fuscus (Anura: Leptodactylidae) out of water: developmental differences and interspecific comparisons. Journal of Herpetology 37:107–115
ROCHA, C. F. D., D. C. PASSOS, AND W. E. MAGNUSSON. 2017. Survival of Amazonian tadpoles under harsh water-limiting conditions. South American Journal of Herpetology 12:212–217.
VALERIO, C. E. 1971. Ability of some tropical tadpoles to survive without water. Copeia 1971:364–365.
VENTURELLI, D.P, Da SILVA, W.R., GIARETTA, A.A. 2021. Tadpoles’ Resistance to Desiccation in Species of Leptodactylus (Anura, Leptodactylidae). Journal of Herpetology, Vol. 55, No. 3, 265–270, 2021

Line 581: There is a metabolic cost associated with producing the foam nest, primarily due to the investment required to manage ammonia toxicity. However, if the tadpoles are clustered together in a moist area (Downie and Smith, 2003), they would still need to cope with ammonia levels, even if they are not actively producing foam. To what extent can we correlate the metabolic cost of foam production with the amount of ammonia excreted?

Line 582: The citation needs to be standardized. Change it to “Belduque-Correa, et al. 2019.”

Line 583: Some species of Leptodactylidae produce trophic eggs during spawning (unfertilized eggs) that the larvae eat within the nest. In addition, the tadpoles can also feed on their dead or weakened siblings (reference below). Did you observe any trophic eggs in the parental foam nest?
PRADO, et al. 2005. Trophic Eggs in the Foam Nests of Leptodactylus Labyrinthicus (Anura, Leptodactylidae): an Experimental Approach. Herpetological Journal, Vol. 15, 279-284.
Da SILVA, W.R.; GIARETTA, A.A. 2009. On the natural history of Leptodactylus syphax with comments on the evolution of reproductive features in the L. pentadactylus species group (Anura, Leptodactylidae). Journal of Natural History, v. 43, n. 3–4, p. 191–203.
Da SILVA, W.R.; GIARETTA, A.A.; FACURE, K.G. 2005. On the natural history of the South American pepper frog, Leptodactylus labyrinthicus (Spix, 1824) (Anura: Leptodactylidae). Journal of Natural History, v. 39, n. 7, p. 555–566.

Line 594: This modulation of nitrogenous compounds in excretion to reduce toxicity is very interesting and has been identified as a survival mechanism for terrestrial tadpoles out of water.
VENTURELLI, D.P, Da SILVA, W.R., GIARETTA, A.A. 2021. Tadpoles’ Resistance to Desiccation in Species of Leptodactylus (Anura, Leptodactylidae). Journal of Herpetology, Vol. 55, No. 3, 265–270, 2021
ALCOCER, L., X. SANTA CRUZ, H. STEINBEISSER, K. H. THIERAUCH, AND E. M. PINO. 1992. Ureotelism as the prevailing mode of nitrogen excretion in larvae of the marsupial frog Gastrothecario bambae (Anura, Hylidae). Comparative Biochemistry and Physiology 101:229–231.
BLACK, J. H. 1974. Larval spade foot survival. Journal of Herpetology 8:371–373
CANDELAS, G. C., AND M. GOMEZ. 1963. Nitrogen excretion in tadpoles of Leptodactylus albilabris and Rana catesbeiana. American Society of Zoologists 203:521–522

Line 605 - 606: Do these larvae then have a greater capacity to tolerate ammonia toxicity?! It would be very interesting to study what mechanisms of ammonia tolerance these are and how they are distributed among anuran species - amazing!

Section: Short-term consequences of extended terrestriality and foam-making
Line 669: Could it be because they were more hydrated than those that were building nests?

Line 705: The citation needs to be standardized. Change it to “Belduque-Correa, et al. 2019.”

Section: Carry-over effects of extended terrestriality and foam-making on development to metamorphosis
Line 755: I believe this work also supports your statement.
RICHTER-BOIX, A. TEJEDO, R, REZENDE, E. 2011. Evolution and plasticity of anuran larval development in response to desiccation. A comparative analysis. Ecology and Evolution.

Conclusion
Ok

References
Include the reference for citation Wilbur & Collins, 1973
Line 956: There is no citation of this article in the text.
Line 985: There is no citation of this article in the text.
Line 1004: There is no citation of this article in the text.
Line 1021: There is no citation of this article in the text.
Line 1059 and 534: Change “Lisondro et al., 2024” to “Lisondro-Arosemena, et al., 2024”.
Line 1104: There is no citation of this article in the text.
Line 1173: There is no citation of this article in the text.
Line 1280: Change “Wells K.D. 2007” to “Wells, K.D. (2007)”

Figure 1: Change “… arrest (C: 4.5 d, 8.5 d and 12.5 d, respectively) to assess their foam-making ability.” To “… arrest (4.5 d, 8.5 d and 12.5 d, respectively) to assess their foam-making ability (C).”

Figure 5 and 6: It might be helpful to include a legend box so that some information won’t need to be repeated on the graph, making it cleaner. Note that Figures A and B have legends, while C and D only show colors and "N," and the others do not provide a color legend. This can be confusing for the reader.

Table 1: Although all data is summarized in a single table, the values are highly disorganized, making it confusing to understand what SD and N represent, especially with the NA data.

Reviewer 3 ·

Basic reporting

The manuscript by Méndez-Narváez and Warkentin examined the effects of prolonged terrestrially and the larval foam-making activity in the larvae of Leptodactylus fragili to assess the short- and long-term consequences on physiological and developmental traits.
This manuscript contains important results but, some formal points should be addressed before publication to allow the readers to understand the scientific background of the present study.
I have listed below some important issues that should be addressed in the paper before its publication; other specific comments are reported in the attached .pdf.

I found substantial disorganization in all sections of the manuscript. The entire paper should be carefully checked to avoid repetition.
- The English language should be improved to ensure that an international audience can clearly understand. Some examples where the language could be improved are reported in the attached .pdf.
- The introduction should be reorganized to focus only on the central question related to the study's goal and should be significantly shortened. Often, there are too many bibliographic references in the sentences, and they should be removed if not needed or if they are too old (for example in lines 70-72, 78 and 87-89).
Furthermore, indications regarding the results of this study should be deleted, as well as the indications of figures (e.g., Fig. 1A, Fig. 1C, Video S1).
- In the materials and methods and in the results sections, the authors often refer to 'tested ages' but in my opinion it would always be appropriate to report the developmental stage (Gosner stage for example) which the authors mention only in some cases.
- The discussion section is a critical point of this work. The first part of the discussion (lines 456-492) can be eliminated/shortened because too many repetitions are already reported in the results or in the introduction sections. The authors must consider that the discussions cannot repeat the obtained results and cannot include references to figures and tables.
Unfortunately, this part represents the great limit of the entire manuscript and should be rewritten entirely.

Please also see the attached .pdf for other specific comments.

Experimental design

no comment

Validity of the findings

no comment

Annotated reviews are not available for download in order to protect the identity of reviewers who chose to remain anonymous.

---

## Round 0.2 · accepted · Accept

Dear authors, I am very pleased to inform you that your article has been accepted for publication in our journal. I hope that you will continue your research on this topic and send us more than one more article as interesting as this one.

·

Basic reporting

No comment. The text has been edited according to the reviewers' comments.

Experimental design

No comment. The text has been edited according to the reviewers' comments.

Validity of the findings

No comment. The text has been edited according to the reviewers' comments.

Additional comments

The authors have carefully addressed all the comments and suggestions provided during the review process. Notably, they made substantial revisions based on one of the reviewers' observations, which significantly improved the quality of the manuscript.
I appreciate the authors' effort in thoughtfully considering and responding to each of my queries.
The manuscript is now well-revised and meets the expected standards for publication.

Reviewer 3 ·

Basic reporting

no comment

Experimental design

no comment

Validity of the findings

no comment

Additional comments

I have carefully read the revised manuscript. The authors have addressed/resolved all significant issues related to the first draft of their manuscript.
The manuscript can now be accepted in its present form.